# Effects of Extended Light/Dark Cycles on Solanaceae Plants

**DOI:** 10.3390/plants13020244

**Published:** 2024-01-15

**Authors:** Tatjana G. Shibaeva, Elena G. Sherudilo, Elena Ikkonen, Alexandra A. Rubaeva, Ilya A. Levkin, Alexander F. Titov

**Affiliations:** 1Institute of Biology, Karelian Research Center, Russian Academy of Sciences, Petrozavodsk 185910, Russia; sherudil@krc.karelia.ru (E.G.S.); likkonen@gmail.com (E.I.); arubaeva@krc.karelia.ru (A.A.R.); levkin-556@mail.ru (I.A.L.); titov@krc.karelia.ru (A.F.T.); 2Institute of Biology, Ecology and Agricultural Technologies, Petrozavodsk State University, Petrozavodsk 185910, Russia

**Keywords:** intermittent light irradiation, light-dark cycle, continuous lighting, tomato, eggplant, pepper, tobacco

## Abstract

The absence of an externally-imposed 24 h light/dark cycle in closed plant production systems allows setting the light environmental parameters in unconventional ways. Innovative lighting modes for energy-saving, high-quality, and yield production are widely discussed. This study aimed to evaluate the effects of the light/dark cycles of 16/8 h (control) and 24/12 h, 48/24 h, 96/48 h, 120/60 h (unconventional cycles) based on the same total light amount, and continuous lighting (360/0 h) on plant performance of some Solanaceae species. Responses of eggplant (*Solanum melongena* L.), sweet pepper (*Capsicum annuum* L.), tobacco (*Nicotiana tabacum* L.), and tomato (*Solanum lycopersicum* L.) plants to extended light/dark cycles and continuous lighting were studied under controlled climate conditions. Plants with two true leaves were exposed to different light/dark cycles for 15 days. Light intensity was 250 µmol m^−2^ s^−1^ PPFD, provided by light-emitting diodes (LEDs). After the experiment, tomato, sweet pepper, and eggplant transplants were planted in a greenhouse and grown under identical conditions of natural photoperiod for the estimation of the after-effect of light treatments on fruit yield. Extended light/dark cycles of 24/12 h, 48/24 h, 96/48 h, 120/60 h, and 360/0 h affected growth, development, photosynthetic pigment content, anthocyanin and flavonoid content, and redox state of plants. Effects varied with plant species and length of light/dark cycles. In some cases, measured parameters improved with increasing light/dark periods despite the same total sum of illumination received by plants. Treatments of tomato and pepper transplants with 48/24 h, 96/48 h, and 120/60 h resulted in higher fruit yield compared to conventional 16/8 h photoperiod. The conclusion was made that extended light/dark cycles can result in increased light use efficiency compared to conventional photoperiod and, therefore, reduced product cost, but for practical application, the effects need to be further explored for individual plant species or even cultivars.

## 1. Introduction

One of the most important innovations of recent decades is plant factories with artificial lighting (PFALs). They are closed-plant production systems based on integrated industrial technologies designed for the year-round production of various crops [1,2]. PFALs have several benefits compared to conventional systems. The major advantages these facilities present are protection from unfavorable weather conditions, increased protection from pests, and control over resource use. They ensure higher product quality and shorter production periods [2]. However, the advantages of PFALs primarily come at the expense of energy due to the need to deliver light energy to plants. Crop production in closed systems requires more energy input than open-air agriculture or regular greenhouse production [3,4]. Therefore, the key challenge of PFALs is both to improve yield and resource use efficiency. Fortunately, closed plant production systems allow for control over many environmental variables that determine plant performance. This control allows for a more predictable plant growth and development rate as well as plant composition, which is an undoubted advantage of PFALs over other farming techniques [5,6].

Among environmental factors that affect plant growth and development, light is one of the most important [7]. Optimizing light conditions is important to maximize the economic benefit of plant production in closed systems [8]. Therefore, research and development supporting PFALs should be aimed at reducing operational costs and the high demand for electrical energy. Improvements in light use efficiency may result in product cost reduction and improve PFAL profitability [6]. The absence of an externally imposed 24-h light/dark cycle in closed plant production systems allows the use of unconventional or abnormal light/dark cycles [9]. There are two factors that characterize the light/dark cycle: the period or the length of the light/dark cycle and the ratio of the illumination time to the dark time. The normal light/dark cycle has a period equal to the diurnal cycle period of 24 h. Otherwise, the light/dark cycle is called an abnormal one [10,11]. So far, related studies have mainly focused on light intensity, photoperiod, daily light integral (DLI), and light quality. There are few reports [9,12,13,14,15,16,17] on the effects of intermittent light, i.e., abnormal light/dark cycles on plant growth and phytochemical content of lettuce, sweet potato, microgreens, *Coffea arabica* and *Dendrobium officinale*. They demonstrated that different abnormal light/dark cycles are beneficial for crop production and have the potential to reduce energy consumption by optimizing yield or other targeted traits. The stimulating effect of unconventional irradiation modes on growth is partially attributed to their positive regulation of photosynthetic efficiency [14]. Besides, intermittent lighting was shown to be capable of enhancing the nutritional value of crops by inducing the accumulation of secondary metabolites [9,15,16]. 

At the same time, recent studies [18,19] have shown that a sudden prolongation of the photoperiod may cause a new form of abiotic stress called photoperiod stress. It results in the accumulation of ROS in the apoplast during the night following an extended light period and a stress response. There is evidence of a photoperiod-dependent regulation of the plant redox state. Photoperiod sensing is linked to redox regulation, allowing efficient light usage and redox balancing as well as preventing oxidative damage when photoperiod changes [20,21]. As the light requirements of plants and their stress responses are subject to species and cultivars, detailed studies of unconventional light/dark cycles on plant growth and development are urgently needed. 

Solanaceae plants are economically important crops in many countries. In this study, we investigated the effects of unconventional light/dark cycles of 24/12, 48/24, 96/48, 120/60, and 360/0 h (continuous lighting) on the growth and development of eggplant, sweet pepper, tobacco, and tomato. The objective of this study was to investigate morpho-physiological responses of young Solanaceae plans to extend light/dark cycles. 

## 2. Results and Discussion

### 2.1. Plant Growth and Development

Table 1 shows that leaf area, leaf number, and leaf mass per area (LMA) values differed among different light/dark cycle treatments for all studied species. Leaf area was generally slightly smaller under abnormal light/dark cycles compared to control treatment (16/8 h) in eggplant, pepper, and tobacco and slightly higher in tomato plants. For all plant species except tobacco, the lowest values were observed under the 120/60 h cycle. In tobacco, the smallest leaves were in plants treated by 48/24 and 96/48 h cycles. LMA increased under all light/dark cycles in eggplant and pepper, while in tobacco and tomato plants, the LMA values exceeded control ones only in plants grown under the 360/0 h cycle. The leaf number was the greatest under continuous lighting (360/0 h) in all plants except tobacco, which had more leaves under the 120/60 h cycle. However, in pepper plants, the differences in leaf number between 16/8 h, 24/12 h, 48/24 h, and 360/0 h cycles were not significant. Other light/dark cycles resulted in inconsistent changes in leaf number with a tendency for leaf number to decrease with increasing the length of the light/dark cycle. Differences in the LMA values and leaf number between plants grown under 24/12 h, 48/24 h, 96/48 h, 120/60 h, and control plants (16/8 h) were smaller than those between continuously irradiated plants (360/0 h) and control plants. This is rather expected as all the treatments except 360/0 h have the same average DLI, which is smaller than the DLI provided by continuous lighting. It also means that differences between the abnormal light/dark cycles, excluding 360/0 h and control treatments, cannot be explained by different amounts of intercepted light and should have another explanation.

### 2.2. Photosynthetic Pigments

Continuous lighting (360/0 h) considerably decreased chlorophyll (Chl) content in all plant species (Table 1). For those highly sensitive to continuous lighting, Solanaceae plants’ Chl content declined by 45–60%. Carotenoid (Car) content in all species except tobacco also considerably decreased under continuous lighting. However, the Car/Chl ratio was higher in all plants treated by continuous lighting. Upon exposure to abnormal light/dark cycles, Chl content decreased in most cases compared to control. However, reductions were less dramatic than under continuous lighting. As to Car content, plant responses were different. The Car content decreased in eggplant, pepper, and tobacco, while the Car content did not change significantly in tomatoes. The Car/Chl ratio increased or tended to increase compared to the control.

### 2.3. Oxidative Stress and Antioxidants

All plants grown under continuous lighting and extended light/dark cycles had higher H_2_O_2_ content compared to control (16/8 h) plants (Table 2). As a general trend, the content of H_2_O_2_ increased with the length of the light/dark cycle, while there was no significant effect of continuous lighting compared to the longest (120/60 h) light/dark cycle. The lipid peroxidation degree determined in terms of malondialdehyde (MDA) content was also higher in plants treated by unconventional light/dark cycles, except for the 24/12 h cycle for tomato and tobacco plants. The content of anthocyanins and flavonoids increased under continuous lighting (360/0 h) essentially in all plant species except eggplant (Table 2). Upon exposure to other abnormal light/dark cycles, the level of anthocyanins and flavonoids increased to a greater or lesser extent in all the plants.

### 2.4. After-Effect of Extended Light/Dark Cycles on Fruit Yield

The fruit yield of plants grown from the transplants that were treated by extended light/dark cycles during the pre-reproductive period varied with plant species. In eggplants, extended light/dark cycle treatments did not have an after-effect on fruit yield, while treatment of transplants by continuous lighting resulted in decreased yield (Figure 1). Pepper and tomato plants grown from transplants treated by 48/24 h, 96/48 h, 120/60 h, and 360/0 h cycles had higher early and total fruit yield compared to control plants. The increase in yield did not correlate with the length of the light/dark cycle.

## 3. Discussion

In this study, we investigated plant responses to extended light/dark cycles that differ from the conventional natural light/dark cycle period of 24 h to which plants are adapted. Plants treated by continuous lighting (360/0 h) demonstrated typical changes aimed at protecting the photosynthetic apparatus against light stress due to an excess of absorbed light beyond that utilized in photosynthesis. Plants grown under continuous lighting had lower Chl content and higher Car/Chl compared to plants grown under a 16/8 h photoperiod. They also had higher H_2_O_2_ content and level of lipid peroxidation accompanied by higher content of such non-enzymatic antioxidants as anthocyanins and flavonoids. As light intensity was the same in all treatments, plants under continuous lighting received greater total light amount compared to plants grown under 16/8 h photoperiod and extended light/dark cycles (average DLI was 21.6 and 14.4 mol m^−2^ day^−1^, correspondingly). Plant responses to continuous lighting could be induced by an increase in DLI or directly result from the continuity of light itself. Continuous light provides a constant flux of energy into photosynthesis, theoretically increasing plant growth and the potential for higher yield [22]. At the same time, continuous lighting alters the physiology of plants by supplying energy components that drive photosynthesis and signaling components perceived by photoreceptors continuously. Thus, continuous lighting may significantly increase photo-oxidative pressure in plants [23]. Therefore, the enhancement of photoprotective mechanisms in these plants in our experiments was rather expected. 

Plants grown under extended light/dark cycles that provided a total light amount equal to that in the control treatment also exhibited an entire spectrum of responses that can be regarded as protective and adaptive ones to excessive illumination. These included all aforementioned changes in the pigment complex as well as increased content of H_2_O_2_ and MDA that evidenced mild oxidative stress. Apparently, photoprotective responses to extended light/dark cycles were induced by long-term (24, 48, 96, and 120 h) light exposure during the first part of the light/dark cycle. Moreover, patterns of light/dark cycles were such that plants could be illuminated during the scotophile phase and vice versa during the photophile phase could concur with darkness. In other words, light was provided to the plants during subjective night, which implies a circadian asynchrony, i.e., the mismatch between the internal (circadian) biorhythms and the external light/dark cycle. All photoperiods longer than 16–18 h would expose plants to light during advanced stages of the scotophile phase (phase of endogenous daily rhythm in which light is inhibitory or has no promoting effect [24]). 

It is known from the recent literature that sudden changes in the photoperiod, in particular its prolongation, cause a photoperiod stress response. It is characterized by the induction of stress response genes, ROS production, and accumulation of jasmonic and salicylic acids [21]. Photoperiod stress signals might have an adaptive value by acting as a priming agent, which improves the plant’s performance under future stress. It was shown that prolongations of the light phase induce a gradually stronger stress response. It indicates that light duration has an impact on the strength of the photoperiod stress response [25]. At the same time, longer prolongations induce true stress (distress), and shorter prolongations of the light period are perceived as not harmful and may present a beneficial stress (eustress) [26]. Despite the fact that Solanaceae plants used in this study are very sensitive to continuous lighting [23,27], no visible damage or inhibition of growth and development has been recorded in plants exposed to extended light/dark cycles.

Beneficial effects of some extended light/dark cycles included higher leaf mass per area, content of anthocyanins and flavonoids, and fruit yield as an after-effect. This corresponds to earlier findings that different light/dark cycle periods affected the energy use efficiency, photosynthesis, and nutritional quality of lettuce [9]. In the study with the same total light amount and electric power consumption in all treatments, the optimal treatment from the perspective of lettuce shoot fresh weight was 24/12 h cycle, while 120/60 h cycle was recommended regarding energy use efficiency and nutritional quality. The assimilation rate of lettuce leaves was about 55% higher than the control under 96/48 h and 120/60 h cycles. There are also findings reported that plant WUE can be regulated by circadian clock genes [28].

Among various environmental factors, light has been found to be one of the most important variables affecting phytochemical concentrations in plants [29,30]. While it is widely understood that light intensity [31,32], photoperiod [32,33,34,35,36,37], and spectral composition of light [32,38,39,40] have significant effects on the nutritional value of plants, this research and few previous studies [9,12,13,14,15,16,17,41] have shown that unconventional light distribution in time as an independent variable may also significantly affect plant nutritional value. Accumulation of substances with antioxidant properties improves plant nutritional value and makes the product beneficial to health and more competitive. Thus, to maximize the economic benefit of obtaining high quality as well as yield in closed production systems, optimizing light distribution in time seems important. 

In this study, extending the light/dark cycle was beneficial for the accumulation of anthocyanins and flavonoids. On the contrary, shortened light/dark cycles (9/3 h, 6/2 h) decreased the total anthocyanin content in lettuce [41]. Owing to their ability to absorb ultraviolet radiation (330–350 nm) and a part of visible rays (520–560 nm), flavonoids protect plant tissues against excess radiation. Anthocyanins, as a subclass of flavonoids, participate in the protection of plant chloroplasts against the high intensity of visible light, acting as a sort of screen absorbing excess photons and, thus, competing with chlorophylls for light energy [42,43,44,45]. This makes it possible to rank them among participants in the nonphotochemical protective mechanism along with the pigments of the xanthophyll cycle. It was noted that anthocyanins perform the photoprotective role more efficiently than carotenoids upon prolonged exposure to light stress, whereas carotenoids of the xanthophyll cycle are more efficient upon brief exposures [46]. However, it is still not known whether photoprotective or antioxidant function is the main for flavonoids and anthocyanins [47,48,49,50]. An important feature of anthocyanins and flavonoids that is significant for humans is their wide range of health-promoting properties, which makes functional food with their elevated content more and more popular [51].

With regards to carotenoids, it is known that, in addition to light-harvesting and photochemical and structural functions, they play a photoprotective role in guarding chlorophyll and other components of photosystems against light overexcitation [52]. Numerous works have shown that elevated content of carotenoids is characteristic of plants under high insolation [53]. However, excessive light does not necessarily elevate the content of carotenoids and sometimes even decreases it to some extent, as shown in this work and earlier [27,37,54]. It is important that under such conditions, the Car/Chl ratio almost always increases, which points to a relatively high concentration of carotenoids within the pool of photosynthetic pigments and promotes the realization of their protective function under excess lighting. 

The key task of PFALs is to reduce costs per product unit. It was shown that the 120/60 h cycle can save at least one-third of the light sources by using movable cultivation beds, thus improving the utilization rate of LEDs and giving full play to the advantages of a long life of LED chips [9]. In our study, extended light/dark cycle treatments of tomato and pepper transplants resulted in after-effects in higher fruit yield. This indicates that these light/dark cycles increased the light use efficiency compared with the conventional 16/8 h photoperiod. 

## 4. Conclusions

Extended light/dark cycles of 24/12 h, 48/24 h, 96/48 h, 120/60 h, and 360/0 h affected the growth, pigment content, and redox state of plants. Effects varied with plant species and length of light/dark cycles. Treatments of tomato and pepper transplants with 48/24 h, 96/48 h, and 120/60 h resulted in higher fruit yield compared to conventional 16/8 h photoperiod. We suggest that extended light/dark cycles may increase light use efficiency compared to conventional photoperiod and, therefore, reduce product cost, but for practical application, the effects need to be further explored for individual plant species or even cultivars.

## 5. Materials and Methods

### 5.1. Plant Material and Growth Conditions

The seeds of eggplant (*Solanum melongena* L.), sweet pepper (*Capsicum annuum* L.), tobacco (*Nicotiana tabacum* L.), and tomato (*Solanum lycopersicum* L.) were germinated for 2 days in Petri dishes on filter paper moistened with distilled water in the darkness at 28 °C. The germinated seeds were planted in individual containers with a commercial soil mixture, and the plants were grown in a Vötsch growth chamber (Burladingen, Germany) at an air temperature of 23 °C and air humidity 70%. Plants were watered daily. 

Light was provided by LEDs (V300 LED Grow Light, ViparSpectra, Richmond, CA, USA) with a light ratio (%) of red:green:blue—50:21:18). The PPFD of 250 μmol m^−2^ s^−1^ was used in all treatments. The PPFD value was measured using LI-250A Light Meter (Li-COR Biosciences, Lincoln, NE, USA).

Six light-dark cycle treatments were set: (1) 16-h light/8-h dark (16/8 h) (control), (2) 24-h light/12-h dark (24/12 h), (3) 48-h light/24-h dark (48/24 h),(4) 96-h light/48-h dark (96/48 h), (5) 120-h light/60-h dark (120/60 h), (6) 360 h light/0 h dark, i.e., continuous lighting during 15 days (360/0 h). These light treatments provided the average daily cumulative intensity of photosynthetically active radiation quanta (daily light integral, DLI) of 14.4 mol m^−2^ day^−1^ for the first five treatments and 21.6 mol m^−2^ day^−1^ for continuous lighting. Plants were treated by these light-dark cycles for 15 days, starting from the two true leaf phases (the 30th day after sowing for tomato and the 45th day after sowing for eggplant, pepper, and tobacco).

After 15 days of growing plants under unconventional light/dark cycles, i.e., at the end of the pre-reproductive period, eggplant, pepper, and tomato transplants were planted in heated plastic film greenhouses (61°47′ N, 34°20′ E). Thereafter, they were grown under natural photoperiod in May–August of 2023 (spring–summer growing season), with all agricultural requirements being fulfilled. The planting density was 3.2 plant m^−2^ for tomatoes and 3.9 plant m^−2^ for eggplants and peppers.

### 5.2. Growth and Yield Measurements

Fully expanded leaves of six plants from each treatment were sampled on the 16th day after the beginning of unconventional light-dark cycle treatments. Leaf area, the number of leaves achieved the length of 10 mm, and more were determined. The values of leaf mass per area (LMA) were calculated as the ratio of a dry mass of the lamina discs to their area. Eight discs were cut from each leaf with an 8-mm diameter cork borer. The dry weight of the discs was determined after their drying to a constant weight at 105 °C.

The early fruit yield (from the first four harvesting dates) and the total fruit yield per square meter were determined for tomato, sweet pepper, and eggplant.

### 5.3. Photosynthetic Pigment Content

The content of Chl*a* and *b* and Car was measured in 96% ethanol extracts with a SF2000 spectrophotometer (Spectrum, St. Petersburg, Russia) and calculated according to the known formulas [55]. The percentage of Chl in light-harvesting complex II (LHCII) was calculated by accepting that almost all Chl*b* is in LHCII and that the ratio of Chl*a* and *b* in LHCII is 1.2 [56].

### 5.4. Anthocyanins and Flavonoids Content

Anthocyanins were extracted from leaves, according to Kang et al. [42]. Fresh leaf tissues (0.1 g) were homogenized in 4 mL of 95%ethanol-1.5 N HCl- (85:15, *v*:*v*). After overnight extraction at 4 °C in darkness, each sample was centrifuged at 10,000× *g* for 5 min. The absorbance of the supernatant was measured at 533 nm (peak of absorption of anthocyanin) and 657 nm (peak of absorption of Chl degradation products). The results were plotted as a difference in absorption at 530 and 657 nm relative to tissue fresh weight (∆A∙g^−1^ FW), and the formula ∆A = A530 − 1/4A657 was used to deduct the absorbance contributed by chlorophyll and its degradation products in the extract [57].

The relative amounts of flavonoids were measured spectrophotometrically [58,59,60]. The supernatant for anthocyanins was diluted10 times and the absorbance was measured at 300 nm. Flavonoids content in the sample was expressed as absorbance at 530 nm g^−1^ fresh weight of tissue.

### 5.5. Malondialdehyde (MDA) Content

The content of malondialdehyde (MDA), the end product of lipid peroxidation, was determined with a standard method based on the reaction of these substances with thiobarbituric acid (TBA) that produces a trimethine complex with an absorption maximum at 532 nm [61]. The value for nonspecific absorption of each sample at 600 nm was also recorded and subtracted from the absorbance recorded at 532 nm. The concentration of MDA was calculated using an extinction coefficient of 155 mM^−1^ cm^−1^. The lipid peroxidation levels were expressed as micromoles of MDA per gram of FW.

### 5.6. Hydrogen Peroxide Content

Hydrogen peroxide content was determined according to Velikova et al. [62]. Leaf tissues (0.1 g) were homogenized in an ice bath with 2 mL 0.1% (*w*/*v*) TCA. The homogenate was centrifuged at 12,000× *g* for 15 min at 4 °C and 0.5 mL of supernatant was added to 0.5 mL potassium phosphate buffer (pH 7.0) and 1 mL 1M KI. The absorbance of the supernatant was measured at 390 nm. The content of H_2_O_2_ was calculated by comparison with a standard calibration curve and expressed in µmol g^−1^ FW.

### 5.7. Data Analysis

Ten plants per treatment were employed in the experiment. The tables and figures show mean values and standard errors. Significant differences between the means were revealed at *p* < 0.05 using the least significant difference test.

## Figures and Tables

**Figure 1 plants-13-00244-f001:**
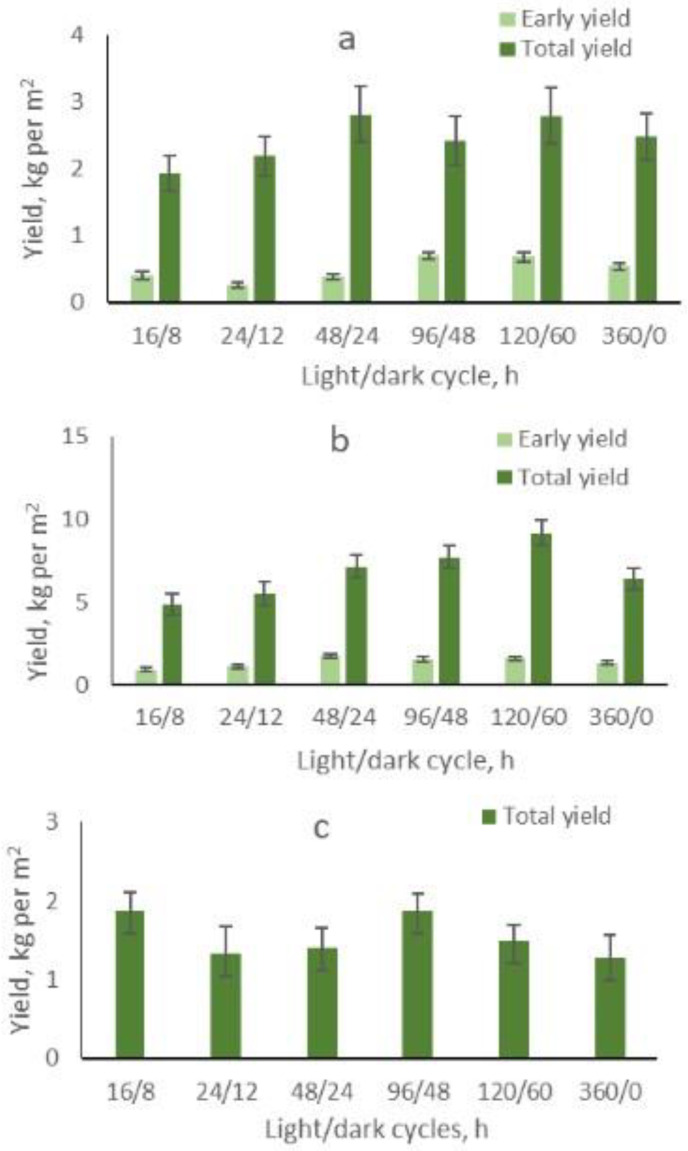
After-effect of pepper (**a**), tomato (**b**), and eggplant (**c**) transplant treatments with extended light/dark cycles on early and total fruit yield.

**Table 1 plants-13-00244-t001:** Leaf area, leaf mass per area, leaf number, chlorophyll, and carotenoid content in plants exposed to different light/dark cycle treatments.

Light/Dark Cycle, h	Leaf Area, dm^2^	LMA, mg cm^−2^	Leaf Number	Chl *a + b*, mg gDW^−1^	Car, mg gDW^−1^
Eggplant					
16/8	8.2 ± 0.8	1.71 ± 0.26	8.3 ± 0.2	17.1 ± 1.8	2.1 ± 0.3
24/12	7.5 ± 0.7	2.22 ± 0.17 *	8.0 ± 0.2	14.7 ± 0.3 *	1.8 ± 0.3
48/24	7.2 ± 0.6 *	2.56 ± 0.43 *	6.9 ± 0.2 *	14.0 ± 2.8 *	1.8 ± 0.1
96/48	6.0 ± 0.6 *	2.30 ± 0.11 *	7.9 ± 0.3	12.3 ± 2.1 *	1.8 ± 0.4
120/60	5.0 ± 0.6 *	2.44 ± 0.17 *	7.1 ± 0.2 *	12.0 ± 0.9 *	1.6 ± 0.3 *
360/0	7.5 ± 0.8	3.22 ± 0.42 *	8.6 ± 0.2 *	8.9 ± 1.0 *	1.1 ± 0.1 *
Pepper					
16/8	6.1 ± 0.6	1.78 ± 0.06	15.3 ± 0.6	19.9 ± 1.0	2.5 ± 0.2
24/12	5.5 ± 0.6	2.09 ± 0.07 *	16.8 ± 0.5	14.5 ± 1.2 *	1.9 ± 0.1 *
48/24	6.3 ± 0.5	2.88 ± 0.25 *	16.0 ± 0.8	12.9 ± 0.8 *	1.6 ± 0.1 *
96/48	6.0 ± 0.6	2.40 ± 0.08 *	13.4 ± 0.8 *	10.9 ± 3.3 *	1.5 ± 0.5 *
120/60	3.5 ± 0.4 *	2.93 ± 0.11 *	14.3 ± 0.4 *	12.7 ± 1.7 *	1.6 ± 0.1 *
360/0	5.5 ± 0.6	2.76 ± 0.12 *	16.0 ± 0.6	9.5 ± 0.6 *	1.4 ± 0.1 *
Tobacco					
16/8	4.5 ± 0.5	3.13 ± 0.30	7.1 ± 0.2	9.9 ± 0.3	1.2 ± 0.1
24/12	6.8 ± 0.6 *	2.90 ± 0.39	7.9 ± 0.3 *	7.5 ± 0.6 *	0.9 ± 0.1 *
48/24	3.4 ± 0.4 *	2.59 ± 0.18	6.7 ± 0.3	4.5 ± 0.6 *	0.5 ± 0.3 *
96/48	2.6 ± 0.3 *	1.94 ± 0.18 *	5.8 ± 0.3 *	6.0 ± 1.1 *	0.8 ± 0.1
120/60	5.4 ± 0.5 *	1.63 ± 0.17 *	7.8 ± 0.3 *	6.5 ± 0.3 *	0.8 ± 0.1 *
360/0	4.4 ± 0.4	5.02 ± 0.28 *	6.6 ± 0.3	5.5 ± 0.3 *	0.6 ± 0.1 *
Tomato					
16/8	8.8 ± 0.7	2.92 ± 0.44	12.0 ± 0.3	19.6 ± 0.3	1.4 ± 0.1
24/12	10.4 ± 0.9 *	2.42 ± 0.45	11.5 ± 0.5	16.2 ± 1.2 *	1.3 ± 0.1
48/24	9.1 ± 0.8	2.75 ± 0.20	12.0 ± 0.2	10.3 ± 1.9 *	1.5 ± 0.1
96/48	9.2 ± 0.9	2.57 ± 0.43	12.1 ± 0.3	13.9 ± 2.0 *	1.4 ± 0.1
120/60	7.5 ± 0.7	2.85 ± 0.38	11.2 ± 0.5	18.2 ± 5.7	1.4 ± 0.1
360/0	12.1 ± 0.9 *	3.88 ± 0.36 *	13.2 ± 0.3 *	7.8 ± 0.5 *	0.9 ± 0.1 *

* indicates significant differences compared with the control (*p* ≤ 0.05).

**Table 2 plants-13-00244-t002:** The malondialdehyde (MDA), H_2_O_2_, anthocyanin, and flavonoid contents of plants exposed to different light/dark cycle treatments.

Light/Dark Cycle, h	H_2_O_2_, mg gDW^−1^	MDA, µmol gDW^−1^	Anthocyanins, (A_530_ − 0.25A_657_) gFW^−1^	Flavonoids, A_300_gFW^−1^
Eggplant				
16/8	0.33 ± 0.04	144.5 ± 10.5	0.54 ± 0.04	22.2 ± 2.1
24/12	0.38 ± 0.03	173.0 ± 13.3 *	0.58 ± 0.11	22.4 ± 0.3
48/24	0.41 ± 0.06 *	163.9 ± 15.0 *	0.53 ± 0.05	37.1 ± 3.5 *
96/48	0.47 ± 0.02 *	189.3 ± 17.4 *	0.55 ± 0.04	40.6 ± 2.7 *
120/60	0.53 ± 0.06 *	163.1 ± 4.7 *	0.51 ± 0.03	33.5 ± 2.9 *
360/0	0.49 ± 0.01 *	172.3 ± 15.5 *	0.50 ± 0.03	26.2 ± 1.1 *
Pepper				
16/8	0.40 ± 0.02	126.9 ± 13.1	0.69 ± 0.04	24.6 ± 1.6
24/12	0.42 ± 0.02	142.0 ± 8.5	0.60 ± 0.08	29.0 ± 3.8 *
48/24	0.44 ± 0.02	127.5 ± 14.0	0.66 ± 0.05	36.6 ± 2.9 *
96/48	0.61 ± 0.02 *	151.0 ± 6.1	0.52 ± 0.06 *	34.0 ± 2.9 *
120/60	0.58 ± 0.06 *	152.5 ± 12.8	0.61 ± 0.06	37.7 ± 3.6 *
360/0	0.47 ± 0.02 *	175.5 ± 13.5 *	0.94 ± 0.02 *	30.8 ± 1.4 *
Tobacco				
16/8	0.23 ± 0.02	138.3 ± 13.0	1.08 ± 0.14	106.5 ±2.8
24/12	0.31 ± 0.05 *	123.6 ± 14.6	1.00 ± 0.08 *	93.5 ± 7.4
48/24	0.32 ± 0.07 *	215.9 ± 22.7 *	0.95 ± 0.21 *	97.3 ± 9.5
96/48	0.26 ± 0.04	213.6 ± 7.0 *	1.07 ± 0.18	87.7 ± 4.0 *
120/60	0.36 ± 0.04 *	200.0 ± 9.0 *	0.90 ± 0.12 *	78.4 ± 9.9 *
360/0	0.31 ± 0.05 *	169.8 ± 25.0 *	1.23 ± 0.09	142.0 ± 3.9 *
Tomato				
16/8	0.47 ± 0.02	164.9 ± 12.1	0.86 ± 0.13	33.9 ± 3.9
24/12	0.62 ± 0.12 *	148.7 ± 18.4	0.87 ± 0.02	34.9 ± 1.3
48/24	0.78 ± 0.11 *	194.6 ± 27.6 *	1.18 ± 0.07 *	34.9 ± 2.0
96/48	0.69 ± 0.06 *	195.1 ± 40.4 *	2.12 ± 0.30 *	41.8 ± 3.8 *
120/60	0.75 ± 0.07 *	250.7 ± 27.5 *	2.15 ± 0.30 *	36.7 ± 3.4 *
360/0	0.85 ± 0.05 *	253.9 ± 24.5 *	1.74 ± 0.15 *	41.6 ± 2.2 *

* indicates significant differences compared with the control (*p* ≤ 0.05).

## Data Availability

The data presented in this study are available on request from the corresponding author.

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
