# Peer review of "Effects of Extended Light/Dark Cycles on Solanaceae Plants"

_plants, 2024, doi:10.3390/plants13020244_

Round 1

Reviewer 1 Report

Comments and Suggestions for Authors

This study entitled “Effects of Extended Light/Dark Cycles on Solanaceae Plants” firstly choses four Solanaceae plants (eggplant, sweet pepper, tobacco, and tomato) as research object. And then according to measures many related indicators such as photosynthetic activity, water use efficiency, photosynthetic pigment content and so on to certify that extending light/dark cycles can result in increased plant production efficiency regarding light consumption. The treatments and workload about this study were rather huge. Meanwhile, the analysis of data is comprehensive. Anyway, I think that this manuscript can be accepted after a major revision in this journal, given that the authors take a note on the comments I have given. Some suggestions for improvement:

(1) There are some formatting and syntax errors that need to be fixed. For example, in line 98 there is an extra space between 0 and extending, and in line 153 "extended" is changed to "extending".

(2) Lines 89-90, conclusion For all plant species the lowest values were obseved under 120/60 h cycle. Not consistent with the tobacco results in Table 1.

(3) Lines 92-94, Conclusion The leaf number was the greatest under continuous lighting (360/0 h) in all plants except tomacco, which had more leaves under the 120/60 h cycle.  The results were inconsistent with those of peppers and tomatoes in Table 1.

(4) Please describe the early stage and total fruit yield in Table 1 in the first part of the conclusion or separate list/graph these two data later.

(5) There are too many tables in this study. Use a combination of charts to illustrate the results.

Comments on the Quality of English Language

The quality of English Language should do moderate editing.

Author Response

We thank the reviewer for the thorough review, supportive comments and constructive critiques on our work. All the reviewer comments and suggestions were considered during the revision and we provide detailed responses below. Corresponding revisions are highlighted in the re-submitted file.

Comments 1: There are some formatting and syntax errors that need to be fixed. For example, in line 98 there is an extra space between 0 and extending, and in line 153 "extended" is changed to "extending".

Response 1: Thank you for pointing this out. Corrections are made.

Comments 2: Lines 89-90, conclusion For all plant species the lowest values were observed under 120/60 h cycle. Not consistent with the tobacco results in Table 1.

Response 2: We agree with this comment. We revised the text on lines 90-91.

Comments 3: Lines 92-94, Conclusion The leaf number was the greatest under continuous lighting (360/0 h) in all plants except tobacco, which had more leaves under the 120/60 h cycle.  The results were inconsistent with those of peppers and tomatoes in Table 1.

Response 3: We agree with this comment. We revised the text on lines 94-96.

Comments 4: Please describe the early stage and total fruit yield in Table 1 in the first part of the conclusion or separate list/graph these two data later.

Comments 5: There are too many tables in this study. Use a combination of charts to illustrate the results.

Response 4, 5: Thank you for the suggestion. We have, accordingly made a separate chart and combined tables 1 and 2.

Reviewer 2 Report

Comments and Suggestions for Authors

Comments paper: “Effects of extended light/dark cycles in Solanaceae plants”

Shibaeva et al., Plants 2773840

The paper describes a study on the use of different lighting modes for energy-saving, high quality and yield production of several members of the Solanaceae family. Five different light/dark cycles indicated as “unconventional cycles” and one continuous light regime were applied. The performance of the plants was evaluated growth parameters such as leaf area, leaf mass per area, leaf number, early and toral fruit yield. Furthermore, physiological parameters including pigment content (chlorophyll – carotenoids), several photosynthetic parameters, indicators of oxidative stress and antioxidants were analysed. From the results, it was concluded that extended light/dark cycles can result in increased plant production efficiency regarding light consumption.

The paper is clearly written and the results, presented in tables, are described in detail. However, there are some issues which have to be considered carefully in order to improve the manuscript.

p.3, line 88: here, “control treatments” is mentioned for the first time. Maybe I missed this, but throughout the whole paper, I could not find any data that represent the control. The control treatment is also not described in Materials & Methods. This is a serious shortfall. In order to interpret the results correctly, the data of the control treatment should added.

On p.5, lines 130-131 and table 3: “Among all species, the highest ratios of intercellular (Ci) to ambient (Ca) CO2 concentration were associated with the highest An. Double check this because there is a discrepancy with the data in the table.

An important parameter should be described clearly: DLI, daily light interval. What is this exactly and why is the DLI the same in the 5 “unconventional cycles”, but different under continuous lighting. See here also in the ‘Conclusion’, p.9, line 280-281: “Our results also showed that an increase  in DLI is not a prerequisite for triggering photoprotective responses in plants leading to the production of antioxidants”. There is only one increase in LDI, in the case of continuous lighting. Make this more clear and understandable for the reader.

 The discussion can also be improved by formulating mechanistic approaches to explain what you wrote as: “extended light/dark cycles can result in increased plant production efficiency regarding light consumption”. The discussion is too descriptive and less explanatory.   

Minor comments:

p.2, line 45: “ptoduction” should be “production”

line 77: “unconbentional” should be “unconventional”

p.3, line 93: “tomacco” should be “tobacco”

Author Response

We thank the reviewer for the thorough review, supportive comments and constructive critiques on our work. All the reviewer comments and suggestions were considered during the revision and we provide detailed responses below. Corresponding revisions are highlighted in the re-submitted file.

Comments 1: p.3, line 88: here, “control treatments” is mentioned for the first time. Maybe I missed this, but throughout the whole paper, I could not find any data that represent the control. The control treatment is also not described in Materials & Methods. This is a serious shortfall. In order to interpret the results correctly, the data of the control treatment should added.

Response 1: Thank you for pointing this out. Plants grown under conventional photoperiod 16/8 h were used as control. It was mentioned in the abstract only (line 15), which is our fault. Therefore, we have added this information to the Materials & Methods and indicated on line 89. Thus, all the data on control treatment are provided.

Comments 2: On p.5, lines 130-131 and table 3: “Among all species, the highest ratios of intercellular (Ci) to ambient (Ca) CO2 concentration were associated with the highest An. Double check this because there is a discrepancy with the data in the table.

Response 2: We agree with this comment. Therefore, we have changed the sentences to ‘While for tomato plants the ratio of intercellular (Ci) to ambient (Ca) CO2 concentration did not vary between light treatments, then for eggplants and peppers the highest Ci to Ca ratios were associated with the 16/8 light regime, and the lowest were found under 24/12 cycle.’ – lines 133-136.

Comments 3: An important parameter should be described clearly: DLI, daily light integral. What is this exactly and why is the DLI the same in the 5 “unconventional cycles”, but different under continuous lighting. See here also in the ‘Conclusion’, p.9, line 280-281: “Our results also showed that an increase  in DLI is not a prerequisite for triggering photoprotective responses in plants leading to the production of antioxidants”. There is only one increase in LDI, in the case of continuous lighting. Make this more clear and understandable for the reader.

Response 3: Daily light integral (DLI) describes the number of photosynthetically active photons (individual particles of light in the 400-700 nm range) that are delivered to a specific area over a 24-hour period. In case of abnormal light/dark cycles it is only possible to use the term average DLI as the length of the cycle differs from 24 h. In order to provide easier understanding we used the term “total light amount” instead of «average DLI» as regarded to abnormal light/dark cycles throughout the text.

Comments 4: The discussion can also be improved by formulating mechanistic approaches to explain what you wrote as: “extended light/dark cycles can result in increased plant production efficiency regarding light consumption”. The discussion is too descriptive and less explanatory.

Response 4: We agree and therefore we have made some changes. Production efficiency is replaced by light use efficiency, which reduces product cost.

Comments 5: p.2, line 45: “ptoduction” should be “production”

Comments 6: line 77: “unconbentional” should be “unconventional”

Comments 7: p.3, line 93: “tomacco” should be “tobacco”

Response 5-7: Thank you for pointing it out. Corrections are made (line 45, 77, 94).

Reviewer 3 Report

Comments and Suggestions for Authors

Comments on the Quality of English Language

Minor editing of English language required

Author Response

We are very grateful to the reviewer for taking time to read our MS, analyze the results and writing a review, reading which showed us deep interest and goodwill. In the introductory part, the reviewer notes that the article presents the results obtained for four crops within the framework of one experiment in Russia. It seemed to us (perhaps mistakenly) that there were two hidden questions in this sentence that still needed to be answered. The first of these can be rephrased as: How representative are the results obtained in one experiment? As is known, representativeness depends not only on the number of repetitions, but also on the object of study, experimental conditions, the degree of variation of the studied indicators and the reliability of the methods used to evaluate them. Considering all these important points, we believe that the data we obtained is quite reliable, reflects the real picture and, therefore, can be trusted. As for the country where the experiment was conducted, although in some studies this may be important and should be taken into account, in our case this circumstance did not play any role. Let us further answer the specific questions that are given in the review.

Comments 1. The first five treatments all used the same amount of energy over the long-term. Yet, the authors mention energy saving. The last treatment (360/0 hours) is confounded with cycling period and total light supply (more than the other treatments).

Response 1. Thank you for pointing this out. Indeed, first five cycles used the same total amount of energy. It means that higher production in any of these treatment means higher light use efficiency or electric use efficiency and therefore reduces product cost, which is important. We do not report lower energy use, but suggest that abnormal light/dark cycles may result in higher resource use efficiency. Moreover, it was shown that in closed production systems with movable cultivation beds extended light/dark cycle 120/60 h saved at least one-third of the light source. The treatment with continuous lighting with higher energy use was taking as contrasting. Photoprotective responses of tomato and eggplant to continuous lighting were expected as they are well documented, but abnormal light/dark cycles with lower total amount of used energy (equal to 16/8 h treatment)  also induced photoprotection mechanisms in plants.

Comments 2. Light levels were low and only 250 µmol/m2/s. How do the yields compare with commercial production under higher light levels?

Response 2: We agree, the light is not high, but PPFD of 250 µmol/(m2 s) cannot be considered as low for Solanaceae plants on vegetative phase. Tomato transplants are often grown under 150-200 µmol/(m2 s), vegetative plants under 200-250 µmol/(m2 s) and fruiting plants under 300-350 µmol/(m2 s). Fruiting plants in our experiments were in the greenhouses under natural light in July-August.

Comments 3. There is a problem with the experimental design and data analyses. The data are not suited to analysis because there were no true replications (no blocks). The authors should note this issue and present means with standard errors or standard deviations.

Response 3. The data in tables and figures are presented as mean values and standard errors.

Comments 4. There was no consistent response to the different light/dark cycles across the species or across the different treatments. Are the results reliable?

Response 4. Other studies also report high species specificity in plant responses to prolonged light/dark cycles. Therefore, we suggest that for practical application the effects need to be further explored for individual plant species or even cultivar.

Comments 5: The data on gas exchange are unreliable and only reflect a single sample (one day?).

Response 5: You are right that this study did not reveal any clear trend of gas exchange under used light regimes. In the section Materials and Methods subsection Net Photosynthesis, Transpiration and Stomatal Conductance we added ‘The parameters were measured on the 16th day after the beginning of light treatments, and not earlier than 2 h after the start of a light period.’

Comments 6. For Table 2, just present data for total chlorophyll and carotenoid. Do these data reflect the different yields in the treatments? Same with the growth data. Do they reflect the yields?

Response 6. We agree that information seems to be excessive, but in some cases, the changes in ratios between pigments, for instance carotenoids to chlorophyll, or chlorophyll in the light-harvesting center better reflect plant photoprotective responses than just their total content. Nevertheless, we removed the data on pigment ratios from the table according to recommendation.

                 Data on pigment content and leaf area or LMA did not directly reflect int the yield assessed in the aftereffect of light/dark treatments. It is obvious that none of the studied indicators is one whose contribution to yield is decisive and highly significant. The yield is an integral indicator, the contribution to which is made by many indicators that characterize the activity of many physiological and biochemical processes in plants (photosynthesis, growth, respiration, water regime, etc.)

Comments 7: What are the authors recommending for commercial production? Are any of the

Response 7: Particular recommendations to commercial production cannot be given yet based on this research. We suggest that extended light/dark cycles, such as 48/24 h, 96/48 h and 120/60 h may increase light use efficiency compared to conventional photoperiod, but for practical application the effects need to be further explored for each plant species or even cultivar. High species specificity in plant responses to prolonged light/dark cycles also reported in other studies.

Comments 8: The ‘Conclusion’ should be reduced to no more than 3-4 sentences highlighting the main findings of the study.

Response 8: Thank you for suggestion. We shortened the conclusion.

Comments 9: The ‘Abstract’ needs to indicate the variable response of yield to the treatments.

Response 9: We agree and indicated it in the abstract. 

Round 2

Reviewer 1 Report

Comments and Suggestions for Authors

According to the revised opinions, the draft is well revised, and the article as a whole has no big problems, and I agree to accept it.

Comments on the Quality of English Language

The English language requires a small amount of modification

Author Response

Dear Reviewer, 

Thank you very much for your suggestions for manuscript improvement. English language has been edited. 

Reviewer 2 Report

Comments and Suggestions for Authors

Dear authors,

Thank you for reviewing the manuscript, most of my comments has been taken into account. I can recommend "accept".

Author Response

Dear Reviewer, 

Thank you very much for your suggestions for manuscript improvement. 

Reviewer 3 Report

Comments and Suggestions for Authors

Plants 2773840

Many thanks to authors for their revision.

I still have some concerns about the manuscript and these are indicated below.

Could the authors kindly check the text to break up some of the exceptionally long (unsuitable) paragraphs.

The data on gas exchange are too unreliable and only relate to a single measurement (on one day) over the whole experiment.  These data should be deleted.

As indicated previously, the data are not suited to analysis of variance (ANOVA) since the treatments were not arranged in blocks/replicates.  The subscripts noting the results of the ANOVAs should be deleted in Tables 1 to 3.

The authors note that the results are variable and more research is required.  This should be indicated at the end of the ‘Abstract’.

In the ‘Conclusion’, indicate the relationship between yield and the light treatments.

The specific effects of the treatments on the yields of the different species should be indicated in the ‘Abstract’.

Comments on the Quality of English Language

See report.

Author Response

We are very grateful to the reviewer for taking time to read our re-submitted MS.

According to your recommendations we (a) modified the text to make it easier to read and understand, (b) deleted the data on gas exchange and subscripts to the tables, and (c) modified the abstract and conclusion.   

Round 3

Reviewer 3 Report

Comments and Suggestions for Authors

Many thanks to the authors for revising the paper.  I recommend that the manuscript be sent to another reviewer for a final check of quality.

Comments on the Quality of English Language

English format is much better in the revision.

Author Response

Dear Reviewer, 

Thank you very much for your valuable suggestions that helped to improve the  manuscript. 
